# Cytotoxic Furan- and Pyrrole-Containing Scalarane Sesterterpenoids Isolated from the Sponge *Scalarispongia* sp.

**DOI:** 10.3390/molecules24050840

**Published:** 2019-02-27

**Authors:** Yeon-Ju Lee, Su Hyun Kim, Hansol Choi, Hyi-Seung Lee, Jong Seok Lee, Hee Jae Shin, Jihoon Lee

**Affiliations:** 1Marine Natural Products Chemistry Laboratory, Korea Institute of Ocean Science and Technology, 385 Haeyangro, Busan 49111, Korea; tngus173@naver.com (S.H.K.); sol8712@kiost.ac.kr (H.C.); hslee@kiost.ac.kr (H.-S.L.); jslee@kiost.ac.kr (J.S.L.); shinhj@kiost.ac.kr (H.J.S.); jihoonlee@kiost.ac.kr (J.L.); 2Department of Marine Biotechnology, University of Science and Technology, 217 Gajungro, Daejeon 34113, Korea

**Keywords:** *Scalarispongia*, sesterterpenoid, furan, pyrrole, conjugate addition, cytotoxicity

## Abstract

Three furan-containing scalarane sesterterpenoids (**1**–**3**) and a novel pyrrole-containing analog (**4**) were isolated from the sponge *Scalarispongia* species. Compound **3**, reported in the literature as a synthetic derivative of furoscalarol **2**, was for the first time isolated from a natural source. During the separation performed using a silica column in the presence of methanol, 16-methoxy derivatives (**5**, **6**) were obtained from the unintended reaction of **2**. The isolated natural products **3** and **4** and the artifact **5** showed moderate to high cytotoxicity against six human cancer cell lines, whereas compound **6**, the C-16 epimer of **5**, showed no cytotoxicity at a concentration of 60 μΜ.

## 1. Introduction

*Scalarispongia* is a genus of sponges that was established in the year 2000; previously, it was assigned as *Cacospongia scalaris* [1]. Marine sponges belonging to this genus are known sources of scalarane sesterterpenoids, which have a characteristic carbon skeleton consisting of four *trans*-fused cyclohexane rings (A/B/C/D) and an optional five-membered heterocycle (E). E rings usually contain oxygen, with a few exceptions that contain nitrogen [2,3,4,5,6,7,8,9]. Scalaranes have been reported to exhibit diverse biological activities, including antimicrobial [10,11,12], antiinflammatory [13,14], and antifeedant activities [15,16]. Moreover, scalaranes often exhibit moderate to significant cytotoxicity [9,17,18,19], although their mechanism of action has not been known for long. A recent study revealed that two scalarane sesterterpenoids showed significant cytotoxicity against cancer cell lines through apoptotic cell death induced by mitochondrial dysfunction and endoplasmic reticulum stresses [20]. The study also confirmed the potential of scalarane compounds as dual inhibitors of topoisomerase II and Hsp90, which are considered as important molecular targets in the development of anticancer drugs. Owing to the biological significance of these molecules, considerable effort has been made to discover or synthesize them [11,21,22,23].

A previous study described the isolation of eight scalarane sesterterpenoids from the extract of *Scalarispongia* sp., which had been collected on the coast of Dokdo, Republic of Korea [24]. Herein, the isolation of additional derivatives containing furan (**1**–**3**) or pyrrole (**4**) is reported. Compound **3** has previously been reported as a synthetic derivative [3], and compound **4** is a novel compound that can be added to the list of the limited number of *N*-heterocycle-containing scalaranes [12,18,19,25]. Additional novel furoscalarol derivatives (**5**, **6**) were obtained by accident during the separation of **2**; these compounds were formed from the reaction between **2** and methanol in the presence of silica. The cytotoxicity of the isolated (**3**, **4**) and synthesized compounds (**5**, **6**) was measured, and the results provide valuable information on the structure-cytotoxicity relationships of scalarane sesterterpenoids.

## 2. Results

### 2.1. Isolation of Scalarane Sesterterpenoids from Scalarispongia sp.

The procedures for sponge collection, extraction, and solvent partitioning were as previously described [24,26]. The obtained dichloromethane-soluble residue was fractionated using ODS resin with aqueous methanol eluent instead of silica with *n*-hexane/ethyl acetate used in the previous study. The ^1^H-NMR analyses revealed that the 10% aqueous methanol fraction contained the substances that had not been separated in the previous investigation. This fraction was separated using HPLC to yield four scalarane natural products (**1**–**4**) and two artifacts (**5**, **6**).

### 2.2. Identification of the Isolated Natural Products *(**1**–**4**)*

Compounds **1** and **2** were obtained separately, although the separation by HPLC using an ODS column with aqueous methanol was incomplete and provided a partial mixture. The comparison of ^1^H- and ^13^C-NMR data with those reported in the literature revealed that **1** and **2** were isoscalarafuran A and furoscalarol, respectively (Figure 1) [3,27].

Compound **3** could be identified as the 16-keto derivative of **2** because an oxymethine signal at δ_H_ 4.70 observed for **2** disappeared in the ^1^H-NMR spectrum, and a ketone signal at δ_C_ 194.6 appeared in the ^13^C-NMR spectrum (Table 1). The HMBC correlations of the proton signals at δ_H_ 2.34 (H-14), 2.47 (H-15β), and 2.57 (H-15α) with the C-16 carbon signal supported this assignment (Figure 2). A NOESY correlation between a proton signal at δ_H_ 5.48 (H-12) and a methyl proton at δ_H_ 1.32 (H-25) indicated the acetoxy group at C-12 being oriented toward the α-face. The molecular formula of C_27_H_38_O_4_ obtained using HR-FAB-MS analysis was in agreement with the above assignment. This compound has been reported only as a synthetic derivative obtained by the oxidation of **2** [3], whereas the 12-deactyl derivative has been isolated from *Cacospongia* species [28].

Judging by the similarity of ^1^H-, ^13^C-, and 2D-NMR spectra, compound **4** was speculated to have the same carbon framework as that of **3** with the only difference being the heteroatom in the E ring (Table 1, Figure 2). Two proton signals at δ_H_ 7.30 (H-19) and 6.32 (H-20), which corresponded to the carbon signals at δ_C_ 119.5 (C-19) and 111.2 (C-20), respectively, in the HSQC spectrum, showed the HMBC correlations with the carbon signals at δ_C_ 121.1 (C-17) and 135.5 (C-18). Based on these observations, the presence of an α,β-disubstituted pyrrole ring was suggested [29,30,31], and a molecular formula C_27_H_39_NO_3_ obtained using HR-FAB-MS supported this assignment.

### 2.3. Chemical Transformation of Furoscalarol *(**2**)* to Its C-16 Methoxy Derivatives *(**5**, **6**)*

As mentioned above, **1** and **2** were not completely separated by HPLC using an ODS column with aqueous methanol and, thus, HPLC using a silica column was attempted. A mixture of dichloromethane and methanol (99.0:1.0) was used as the mobile phase to separate the mixture of **1** and **2** (1.00:1.50). Consequently, the mixture of **1**, **2**, and two additional derivatives was obtained in an approximate ratio of 1.0:0.5:0.9:0.3 (**1**:**2**:**5**:**6**), calculated based on the integration of ^1^H-NMR spectra (Appendix A). The artifacts **5** and **6** were separated from the mixture by HPLC using an ODS column with aqueous acetonitrile.

The ^1^H- and ^13^C-NMR spectra for both **5** and **6** were considerably similar to those of furoscalarol (**2**), except for the presence of a methoxy signal (**5**, δ_H_ 3.40, δ_C_ 56.6; **6**, δ_H_ 3.44, δ_C_ 56.9; Table 2). The HMBC spectrum measured for **5** showed correlations between the oxymethine proton (H-16) at δ_H_ 4.18 and the carbon signals at δ_C_ 45.9 (C-14), 116.4 (C-17), and 158.9 (C-18) suggesting that the methoxy group was attached to C-16 (Figure 3). Considering the same molecular formula of C_28_H_42_O_4_ obtained using HR-FAB-MS analysis and the similarity of ^1^H- and ^13^C-NMR spectra, compounds **5** and **6** could be verified as a pair of C-16 epimers, as depicted in Figure 1.

A NOESY experiment was performed to verify the stereochemistry at C-16 of **5**, but neither the pseudoequatorial proton at C-16 nor the methoxy proton exhibited correlations that would provide useful information. Alternatively, we compared the coupling pattern of H-16 of **5** and **6** with those of the reported furanoscalarane derivatives bearing an oxymethine at C-16. The oxymethine proton signal of **5** appeared at δ_H_ 4.18 (H-16) as a doublet with a coupling constant of 3.5 Hz, whereas that of **6** was observed at δ_H_ 4.34 (H-16) as a doublet of doublets with coupling constants of 8.5 and 7.0 Hz. The oxymethine signal of **2** appeared at δ_H_ 4.70 (H-16) as a doublet of doublets with *J* = 10.0 and 6.5 Hz, indicating that the pseudoaxial hydrogen was the one with the α-orientation [3]. Therefore, the 16-methoxy group of **5** was speculated to be oriented toward the α-face, and that of **6** was thought to have a β-orientation. This assignment is in accordance with the findings of previous studies that described the structures of furan-containing scalaranes. In the case of the furoscalarol derivative, the oxymethine proton at C-16 appeared as a doublet of doublets with *J* = 4.1 and 1.9 Hz when the attached acetoxy group was oriented toward the α-face [32], whereas that of the β-acetoxy derivative appeared as a doublet of doublets with *J* = 9.4 and 6.6 Hz [7]. The same tendency was observed with the scalarafuran derivatives. Scalarafuran bearing a 16-β-acetoxy group exhibited the related oxymethine signal as a triplet of doublets with coupling constants of 8.5 and 1.5 Hz [33], whereas its C-16 epimer exhibited this signal as a triplet with a coupling constant of 2.7 Hz [19]. Epimeric isoscalarafuran A (**1**) and B also showed this tendency [27].

The reproducibility of the reaction occurring in the HPLC column was verified by treating the mixture of **1** and **2** (1.5:1.0) in dichloromethane/methanol (99:1) with silica and stirring in a flask at ambient temperature for 60 min. After silica was removed by filtration, the resulting residue was concentrated and analyzed using ^1^H-NMR spectroscopy, and the ratio of **1**, **2**, **5**, and **6** was found to be 1.0:0.1:1.2:0.3 (Appendix A). The combined yield of **5** and **6** based on the recovery of the starting material was 78%, suggesting that the degradation of **2** had occurred collaterally during the reaction.

### 2.4. Cytotoxicity Evaluation of the Obtained Compounds

The isolated and synthesized compounds were screened for cytotoxicity against six human cancer cell lines (Table 3). The activity of **1** and **2** could not be measured because of their degradation. The cytotoxicity was found to be reduced by around one-third against all the investigated cell lines when a furan was substituted with a pyrrole. The change in stereochemistry at C-16 remarkably altered the cytotoxicity of the compounds. Compound **5** showed cytotoxicity with GI_50_ values of 7.3 to 8.8 μM against the tested cell lines, whereas **6** exhibited no cytotoxicity at a concentration of 60 μΜ.

## 3. Discussion

Three furan-containing scalarane sesterterpenoids (**1**–**3**) and a novel pyrrole-containing derivative (**4**) were isolated from the *Scalarispongia* sponge extract. To our knowledge, this is the first report of the discovery of compound **3** from a natural source.

Two additional derivatives (**5**, **6**) were obtained through the unintended chemical transformation of furoscalarol (**2**). It could be postulated that the slightly acidic character of silica facilitated the formation of oxocarbenium, which was followed by nucleophilic conjugate addition of methanol (Figure 4). The formation of an α-product (**5**) was likely favored because of the approach of the nucleophile toward the pseudoaxial direction for maximum overlap with the *p*-orbital.

The cytotoxicity evaluation of the obtained compounds showed a remarkable decrease in the activity of **6** compared to that of **5**. A similar tendency was found in the case of 16-acetoxy derivatives of **2**. The 16-α-acetoxy derivative of **2** was reported to inhibit the growth of cancer cell lines such as P388, A549, HT-29, and MEL-28 with GI_50_ values of 1.0 μg/mL [32]. In another report by the same authors, the 16-β-acetoxy derivative was shown to exhibit lower cytotoxicity against the same cell lines with GI_50_ values in the range of 2.5 to 10.0 μg/mL [7].

It is clear that the stereochemistry at C-16 affects the cytotoxicity of furanoscalarane compounds; however, a further literature review suggested that the effects are very complex to be interpreted as a single trend. In the case of sesterstatin 4 and 5, which are the 12-hydroxy derivatives of **1**, the activity of each compound against a panel of cancer cell lines was reported to be very contrastive [34]. Sesterstatin 4, bearing a 16-α-hydroxy group, was cytotoxic against KAT-4 and SW1736 cell lines (GI_50_ 2.0 and 2.1 μg/mL, respectively), whereas sesterstatin 5, a 16-β-hydroxy derivative, showed no cytotoxicity against these cell lines. The opposite tendency was observed with RPMI-7951 and U251 cell lines, whereas both compounds showed similar GI_50_ values ranging from 1.6 to 2.5 μg/mL against BXPC-3, NCI-H460, FADU, and DU145 cell lines. Further studies are required to understand the mode of the cytotoxic action of furanoscalaranes for explaining the reason for this complexity.

## 4. Materials and Methods

### 4.1. General Experimental Procedure

HPLC was performed using the YMC-Pack Silica or YMC-Pack Pro C18 columns and a Shodex RI-101 detector (Showa Denko K. K., Tokyo, Japan). ^1^H-NMR spectra were recorded using a Varian Unity 500 (500-MHz) spectrometer (Varian Inc., Palo Alto, CA, USA). Chemical shifts are reported in ppm from tetramethylsilane, used as the internal references (CDCl_3_: δ_H_ 7.26 ppm), with solvent resonance resulting from incomplete deuteration. ^13^C-NMR spectra were recorded using a Varian Unity 500 (125-MHz) spectrometer with complete proton decoupling (Varian Inc., Palo Alto, CA, USA). Chemical shifts are reported in ppm from tetramethylsilane with the solvent as the internal reference (CDCl_3_: δ_C_ 77.26 ppm). High-resolution mass spectra (HRMS) were obtained with a JEOL JMS-700 spectrometer at the Korea Basic Science Institute (JEOL Ltd., Tokyo, Japan). The optical rotations were measured using a JASCO digital polarimeter (JASCO International Co. Ltd., Tokyo, Japan), using a 5-cm cell. IR spectra were recorded using a JASCO FT/IR-4100 (JASCO International Co. Ltd., Tokyo, Japan). UV spectra were obtained using a Shimadzu UV-1650PC spectrophotometer (Shimadzu Corporation, Kyoto, Japan).

### 4.2. Collection and Extraction of Biological Material

The sponge *Scalarispongia* sp. was harvested by hand upon SCUBA diving at a 10-m depth, offshore of Dokdo (island), Republic of Korea. This sponge was extracted as previously reported [24,26].

### 4.3. General Experimental Protocol for the Isolation and Identification of Compounds

The extract (5.4 g) was partitioned between *n*-butanol and water, and the organic layer (1.8 g) was further partitioned between 15% aqueous methanol and *n*-hexane. The aqueous methanol fraction (1.1 g) was subjected to reverse-phase column chromatography (YMC Gel ODS-A, 60Å, 230 mesh) with a step-up gradient solvent system of 30%, 50%, 70%, 90%, and 100% MeOH/H_2_O. The fraction eluted with 90% MeOH/H_2_O (80.0 mg) was subjected to reverse-phase HPLC using an ODS column with aqueous methanol, to yield **3** (4.3 mg), **4** (6.5 mg), **1** (12.0 mg), and **2** (2.0 mg), along with a mixture of **1** and **2** (1.0:1.5, 18.0 mg). The mixture of **1** and **2** was subjected to HPLC using a silica column to yield a mixture of **1**, **2**, **5**, and **6** (1.0:0.5:0.9:0.3, 16.3 mg), which was further resolved via reverse-phase HPLC with aqueous acetonitrile to yield **1** (5.8 mg), **2** (2.6 mg), **5** (6.0 mg), and **6** (1.7 mg). Known compounds (**1**, **2**) were identified through comparison of ^1^H- and ^13^C-NMR spectra and the HRMS data with those reported previously.

Compound **3**: a pale yellow amorphous solid; ${\left[\alpha \right]}_{D}^{25}$ 36.0 (*c* 0.3, CHCl_3_); UV λ_max_ (log *ε*) 254 (3.84) nm; IR (KBr) ν_max_ 2927, 1742, 1681, 1462, 1372, 1236, 1046 cm^−1^; ^1^H- and ^13^C-NMR (CDCl_3_, 500 and 125 MHz), see Table 1; (+)-HRFABMS *m/z* 427.2846 [M + H]^+^ (calcd for C_2__7_H_3__9_O_4_, 427.2848).

Compound **4**: a pale yellow amorphous solid; ${\left[\alpha \right]}_{D}^{25}$ 35.3 (*c* 0.3, CHCl_3_); UV λ_max_ (log *ε*) 255 (2.98) nm; IR (KBr) ν_max_ 3330, 2926, 1733, 1643, 1462, 1370, 1262, 1027 cm^−1^; ^1^H and ^13^C-NMR (CDCl_3_, 500 and 125 MHz), see Table 1; (+)-HRFABMS *m/z* 426.3011 [M + H]^+^ (calcd for C_2__7_H_3__9_NO_3_, 426.3008).

Compound **5**: a pale yellow amorphous solid; ${\left[\alpha \right]}_{D}^{25}$ −42.4 (*c* 0.3, CHCl_3_); UV λ_max_ (log *ε*) 218 (4.05) nm; IR (KBr) ν_max_ 2957, 1766, 1463, 1373, 1058 cm^−1^; ^1^H- and ^13^C-NMR (CDCl_3_, 500 and 125 MHz), see Table 2; (+)-HRFABMS *m/z* 443.3129 [M + H]^+^ (calcd for C_2__8_H_43_O_4_, 443.3161), *m/z* 411.2901 [M-OCH_3_]^+^ (calcd for C_27_H_39_O_3_, 411.2899).

Compound **6**: a pale yellow amorphous solid; ${\left[\alpha \right]}_{D}^{25}$ 11.6 (*c* 0.3, CHCl_3_); UV λ_max_ (log *ε*) 219 (4.32) nm; IR (KBr) ν_max_ 2957, 1766, 1463, 1373, 1058 cm^−1^; ^1^H- and ^13^C-NMR (CDCl_3_, 500 and 125 MHz), see Table 2; (+)-HRFABMS *m/z* 443.3129 [M + H]^+^ (calcd for C_2__8_H_43_O_4_, 443.3161), *m/z* 411.2901 [M-OCH_3_]^+^ (calcd for C_27_H_39_O_3_, 411.2899).

### 4.4. Cytotoxicity Assay

Six human cancer cell lines, particularly HCT-15 (colon, ATCC CCL-225), NCI-H23 (lung, ATCC CRL-5800), ACHN (renal, ATCC CRL-1611), MDA-MB-231 (breast, ATCC HTB-26), NUGC-3 (stomach, JCBRB 0822), and PC-3 (prostate, ATCC CRL-1435), were used in this study. The cell lines were cultured in RPMI 1640 supplemented with 10% fetal bovine serum. Cell cultures were incubated at 37 °C under a humidified atmosphere of 5% CO_2_. The growth inhibition assays were performed as reported previously [35]. In brief, cancer cells were seeded in a 96-well plate containing control (doxorubicin) or test compounds. After a 48-h incubation, cultures were fixed with 50% trichloroacetic acid (50 μg/mL) and stained with 0.4% sulforhodamine B in 1% acetic acid. Unbound dye was eliminated by washing with 1% acetic acid and protein-bound dye was extracted with 10 mM Tris base (pH 10.5), and the optical density was determined at 540 nm using a VersaMax microplate reader (Molecular Devices, LLC., San Jose, CA, USA). GI_50_ values were calculated using GraphPad Prism 4.0 software (GraphPad Software, Inc., San Diego, CA, USA).

## Figures and Tables

**Figure 1 molecules-24-00840-f001:**
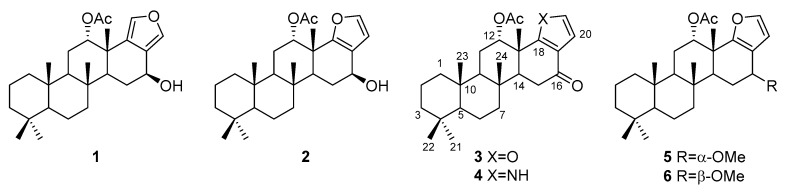
Structures of the obtained furan- and pyrrole-containing scalarane analogs.

**Figure 2 molecules-24-00840-f002:**
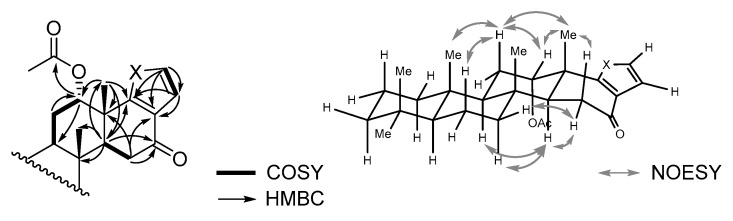
Selected common COSY, HMBC, and NOESY correlations for compounds **3** and **4**.

**Figure 3 molecules-24-00840-f003:**
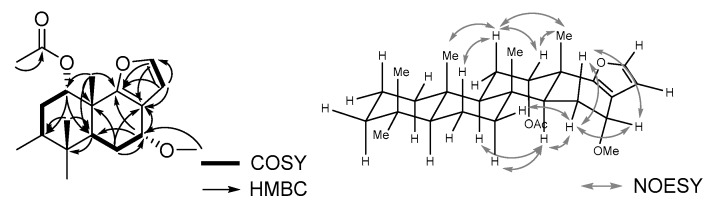
Selected COSY, HMBC, and NOESY correlations for compound **5**.

**Figure 4 molecules-24-00840-f004:**
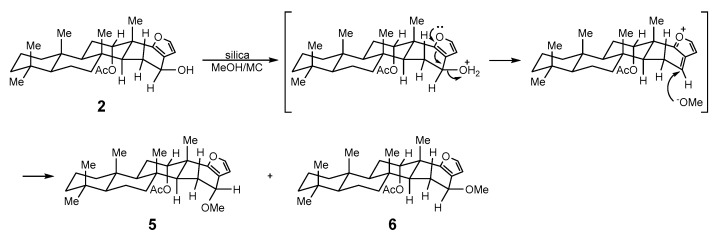
Chemical transformation of **2** into **5** and **6**.

**Table 1 molecules-24-00840-t001:** ^1^H- and ^13^C-NMR data (500 and 125 MHz) for compounds **3** and **4**
^a, b^.

Position	3			4		
	δ_C_, Type ^b^		δ_H_ (*J* in Hz)	δ_C_, Type ^b^		δ_H_ (*J* in Hz)
1	39.9	CH_2_	0.62, br dd (14.0, 14.0)	39.9	CH_2_	0.63, ddd (14.5, 14.5, 4.5)
			1.58, m			1.58, m
2	18.7	CH_2_	1.38, m	18.3	CH_2_	1.39, m
			1.61, m			1.56, m
3	42.1	CH_2_	1.13, m	42.2	CH_2_	1.12, ddd (13.0, 13.0, 4.0)
			1.36, m			1.35, m
4	33.5	C		33.5	C	
5	56.8	CH	0.83, m	56.7	CH	0.83, m
6	18.2	CH_2_	1.31, m	18.7	CH_2_	1.40, m
			1.57, m			1.58, m
7	41.2	CH_2_	1.06, br dd (13.5,10.5)	41.4	CH_2_	1.04, ddd (11.5,11.5,3.5)
			1.78, m			1.77, m
8	37.7	C		38.0	C	
9	53.1	CH	1.31, m	52.7	CH	1.38, m
10	37.2	C		37.2	C	
11	21.9	CH_2_	1.81, m	22.6	CH_2_	1.63, m
			1.92, m			1.81, m
12	73.0	CH	5.48, br s	74.8	CH	5.45, br s
13	41.7	C		39.5	C	
14	50.9	CH	2.34, br d (13.5)	51.6	CH	2.19, dd (10.0, 7.3)
15	34.9	CH_2_	2.47, dd (17.5,13.5)	35.4	CH_2_	2.48, br d (7.3)
			2.57, br d (17.5)			2.49, br d (10.0)
16	194.6	C		196.8	C	
17	120.2	C		121.1	C	
18	172.7	C		135.5	C	
19	142.9	CH	7.26, br s	119.5	CH	7.30, br s
20	106.5	CH	6.59, br s	111.2	CH	6.32, br s
21	33.5	CH_3_	0.85, s	33.5	CH_3_	0.84, s
22	21.5	CH_3_	0.81, s	21.6	CH_3_	0.81, s
23	16.3	CH_3_	0.85, s	16.4	CH_3_	0.84, s
24	17.0	CH_3_	0.99, s	17.1	CH_3_	0.98, s
25	20.4	CH_3_	1.32, s	25.2	CH_3_	1.27, s
12-OAc	172.7	C		171.2	C	
	21.4	CH_3_	1.88, s	21.5	CH_3_	1.91, s

^a^ Data were obtained in CDCl_3_. ^b^ The assignments are based on HSQC, COSY, and HMBC results.

**Table 2 molecules-24-00840-t002:** ^1^H- and ^13^C-NMR data (500 and 125 MHz) for compounds **5** and **6**
^a^.

Position	5			6		
	δ_C_, Type ^b^		δ_H_ (*J* in Hz)	δ_C_, Type ^c^		δ_H_ (*J* in Hz)
1	39.9	CH_2_	0.62, ddd (13.0, 13.0, 3.0)	39.9	CH_2_	0.61, br dd (12.0, 12.0)
			1.57, m			1.57, m
2	18.7	CH_2_	1.40, m	18.4	CH_2_	1.40, m
			1.60, m			1.60, m
3	42.2	CH_2_	1.11, br dd (12.5, 5.0)	42.2	CH_2_	1.12, m
			1.35, m			1.35, m
4	33.5	C		33.5	C	
5	56.5	CH	0.90, m	55.8	CH	0.90, m
6	18.3	CH_2_	1.44, m	18.7	CH_2_	1.40, m
			1.60, m			1.57, m
7	41.4	CH_2_	1.11, m	41.6	CH_2_	1.04, ddd (12.0, 12.0, 4.5)
			1.82, m			1.88, m
8	37.2	C		37.5	C	
9	53.2	CH	1.35, m	53.4	CH	1.24, m
10	37.2	C		37.2	C	
11	22.0	CH_2_	1.74, br d (14.0)	21.9	CH_2_	1.75, m
			1.82, m			1.83, m
12	73.6	CH	5.42, br s	73.8	CH	5.40, br s
13	41.2	C		40.9	C	
14	45.9	CH	2.14, br d (13.0)	49.9	CH	1.73, m
15	23.8	CH_2_	1.59, m	24.8	CH_2_	1.48, m
			2.07, br d (13.0)			2.20, br dd (11.0, 5.0)
16	71.8	CH	4.18, d (3.5)	75.3	CH	4.34, dd (8.5, 7.0)
17	116.4	C		118.2	C	
18	158.9	C		157.6	C	
19	141.0	CH	7.19, br s	141.4	CH	7.18, br s
20	110.2	CH	6.28, br s	108.7	CH	6.30, br s
21	33.5	CH_3_	0.85, s	33.5	CH_3_	0.85, s
22	21.5	CH_3_	0.81, s	21.6	CH_3_	0.82, s
23	16.2	CH_3_	0.83, s	16.2	CH_3_	0.83, s
24	17.6	CH_3_	0.93, s	17.4	CH_3_	0.94, s
25	21.0	CH_3_	1.18, s	22.3	CH_3_	1.28, s
12OAc	170.8	C		170.5	C	
	21.6	CH_3_	1.86, s	21.4	CH_3_	1.84, s
16OMe	56.6	CH_3_	3.40, s	56.9	CH_3_	3.44, s

^a^ Data were obtained in CDCl_3_. ^b^ The assignments are based on HSQC, COSY, and HMBC results. ^c^ Carbons correlating with the corresponding proton in HSQC analysis.

**Table 3 molecules-24-00840-t003:** Growth inhibition by compounds **3**–**6** against a panel of human tumor cell lines ^a^.

Compound	Cell Line (GI_50_ μM) ^b^					
	HCT-15	NCI-H23	ACHN	MDA-MB-231	NUGC-3	PC-3
**3**	8.2	7.1	8.0	7.3	6.5	8.1
**4**	25.0	26.2	26.2	23.2	14.9	24.7
**5**	8.1	7.8	7.4	7.3	7.9	8.8
**6**	>60	>60	>60	>60	>60	>60
**Doxorubicin**	0.2	0.1	0.1	0.1	0.1	0.2

^a^ HCT-15, colon cancer; NCI-H23, lung cancer; ACHN, renal cancer; MDA-MB-231, breast cancer; NUGC-3, stomach cancer; PC-3, prostate cancer. ^b^ Data are an average of at least two tests.

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
