# Peer review of "Cytotoxic Furan- and Pyrrole-Containing Scalarane Sesterterpenoids Isolated from the Sponge Scalarispongia sp."

_molecules, 2019, doi:10.3390/molecules24050840_

Round 1

Reviewer 1 Report

This manuscript reports the structural determination and the cytotoxicity of furan- and pyrrole containing scalarane sesterterpenoids (1-4) from the sponge Scalarispongia species. The author also reported the unexpected side reaction of 2 during the silica column purification, which resulted in the methoxy substituted product 5 and 6. NMR and MS analyses were reliable to determine those structures. The cytotoxic analysis would be helpful information to understand the structure−cytotoxicity relationships of scalarane sesterterpenoids. Also, pyrrole containing analogue 4 could be a new member of the limited number of N-heterocycle-containing scalaranes. This work seems to provide a certain contribution to the related research field, however, revisions to the paper are required in the followings.  

1.     Throughout the whole of this paper, the author should give more comments about 2D NMR analyses, which are determinant to identify the structures. For example, the NOESY correlation for 5 and 6 to conclude the orientation of methoxy group.

2.     In line 78, the α,β-disubstituted pyrrole motif was speculated from the chemical shift of 13C NMR. Providing a reference about this point is helpful for reader to understand.

3.     In line 146, the author mentioned the steric hindrance of C-13 methyl group as one of the facter for α-product 5 preference. It seems to be located far from reaction site C-16. Any supporting data should be provided. 

4.     In line 157, the bold letter sesterstatin and 5should be replaced to not bold representation, sesterstatin 4 and 5.

Author Response

Thank you for your valuable advice on this manuscript. We have tried to improve this manuscript by reflecting your comments.

More comments on 2D NMR analyses are given in the revised version (L67– L73, L76–L82, L103–L106).

Regarding the NMR shifts of α,β-disubstituted pyrrole, three references were added (L81).

We have deleted the comments on the effect of the C-13 methyl group on the diastereoselectivity of the reaction of 2 because it was a controversial issue inside us, and I agree with your opinion that the distance between C-13 and the reaction site is too long (L149).

We have corrected the bold letter as you have pointed out (L160).

We have considered other reviewers’ comments as well to improve the manuscript. We hope that the revised manuscript would be qualified enough for publication.

Reviewer 2 Report

This paper presents the results from the extraction of a sponge found off the coast of S. Korea and identifies three previously described compounds as well as a novel nitrogen containing compound that had been previously synthesized. The authors also present the presence of two artifacts created during the separation of two of the previously described compounds which represent new molecular entities in the same general class. The compounds are well described in the text with appropriate support. The NOESY and COSY spectra however in the supplemental materials should be either re-run or removed as the threshold appears to have been set too low. I do not think however that those spectra are necessary to be included as the correlations are given in complete tables between all the techniques used to identify the compounds. 

The tables are clearly presented. The techniques used to extract the compounds from the sponges were previously described, and modifications included in this paper. 

Cell culture results are from a panel of 6 cell lines. The authors cited a previous method of conducting the cell culture experiments but they need to include the source of the cells and the identifying information such as if they were obtained from ATCC, then the ATCC number must be included in addition to these more common designations. They should also include the growth media for these cells. In other words, the previous method they cite is from 1994, and updating the method should be included in the methods section. 

I am amused that the authors chose to use the term "by accident" when referring to the productions of the two artifact compounds. 

This is a sound paper and should be published after making the adjustments suggested above. 

Author Response

Thank you for your valuable advice on this manuscript. We have tried to improve this manuscript by reflecting your comments.

Instead of removing the NOESY and COSY spectra in the supplementary material, we have chosen to increase the threshold of each spectrum for clear presentation. We believe that the revised supplementary material would look much better than before (Supplementary material).

Regarding the cytotoxicity evaluation method, we have added ATCC number and growth condition including media of the cell lines, and the brief assay protocol (L212–223).

We have considered other reviewers’ comments as well to improve the manuscript. We hope that the revised manuscript would be qualified enough for publication.

Reviewer 3 Report

This is a very well written and presented paper that shows thorough scientific work. I recommend its acceptance.

Some very minor English suggestions and corrections can be found in attach.

Manuscript: molecules 453219
Cytotoxic Furan- and Pyrrole-containing Scalarane 2 Sesterterpenoids Isolated from the Sponge 3 Scalarispongia sp. 4
Yeon-Ju Lee 1, 2,*, Su Hyun Kim 1, 2, Hansol Choi 1, Hyi-Seung Lee 1, 2, Jong Seok Lee 1, 2, Hee Jae 5 Shin 1, 2, and Jihoon Lee 1,
English suggestions:
Instead of ‘---‘ – use ‘---'
Line 17 ‘synthesized’ – ‘synthetic’
Line 17 ‘was first discovered from its natural source’ – is for the first time isolated from a natural source’
Line 20 ‘a synthesized derivative 5’ – the artifact 5
Line 43 ‘synthesized’ – ‘synthetic’
Line 45 ‘ by accident’ – ‘as artifacts’
Line 69 - ‘synthesized’ – ‘synthetic’
Line 70 - ‘synthesized’ – ‘synthetic’
Line 88 – ‘derivatives was obtained’ – derivatives were obtained’
Line 140 – ‘ its natural source’ – ‘a natural source’
1. Introduction
Line 42 and 43 - Insert reference [3] for compound (3).
2. Results
2.1 – Isolation
No corrections
2.2 – compounds 1-4
No corrections

2.3 – Chemical transformation
No corrections
2.4 – cytotoxicity
No corrections
3. Discussion
No corrections
4. Materials and Methods
No corrections
5. References
No corrections

Author Response

Thank you for your valuable advice on this manuscript. We have tried to improve this manuscript by reflecting your comments. Most of your modifications were reflected (L17, 20, 43, 72, 144), however, we have decided to use the term ‘by accident’ because one of the other reviewers mentioned that he(she) was amused by us using that term.

We have considered other reviewers’ comments as well to improve the manuscript. We hope that the revised manuscript would be qualified enough for publication.

Round 2

Reviewer 1 Report

This reviewer thinks that the authors precisely responded to the reviewers' comments.